# Changes in Athletic Performance in Children Attending a Secondary School with a Physical Activity Profile

**DOI:** 10.3390/sports10050071

**Published:** 2022-05-02

**Authors:** Tomas Peterson, Staffan Ek, Ola Thorsson, Magnus K. Karlsson, Magnus Dencker, Per Wollmer

**Affiliations:** 1Department of Sport Sciences, Malmö University, 205 02 Malmö, Sweden; 2Department of Translational Medicine, Lund University, 221 00 Lund, Sweden; staffanek@gmail.com (S.E.); ola.thorsson@med.lu.se (O.T.); magnus.dencker@skane.se (M.D.); per.wollmer@med.lu.se (P.W.); 3Department of Medical Imaging and Physiology, Skåne University Hospital, 205 02 Malmö, Sweden; 4Clinical and Molecular Osteoporosis Research Unit, Department of Clinical Sciences and Orthopaedics, Lund University, Skåne University Hospital, 205 02 Malmö, Sweden; magnus.karlsson@med.lu.se

**Keywords:** sports school, talent identification, longitudinal, elite sports, multidisciplinary

## Abstract

The longitudinal and multidisciplinary research project Malmö Youth Sport Study measured the sports results achieved by two cohorts of pupils using a variable named ACHIEVE, dividing the pupils into three categories (an elite group competing at the national or international level, a group competing at the district level, and a third group either not competing at all or below district level). This was assessed three and six years after baseline at age 13. An additional hypothetical measure, based on information from the athletes’ trainers, predicted the category the pupils were expected to belong to after twelve years (age 25). Social variables related to the ACHIEVE variable are sex, socio-economic position of the parents, ethnicity, completed secondary sports school, sports capital, and quartile of birth. After three years, 28% of the pupils belonged to the elite group and after six years, 26%. Thirty-two and 48%, respectively, had abandoned their elite efforts. The elite group remained fairly stable over time but fewer girls than boys advanced to the elite group. The pupils at the school have a homogenous middle-class background. We found little evidence that socio-economic factors affected ACHIEVE. Nearly all parents had been engaged in sports, either competing or as coaches. On admission to the school, there was a pronounced relative age effect (RAE). This remained after three years as the age was significantly different between the three groups but was reduced after six years. According to the prognosis made by the coaches, the elite group would be considerably smaller when the subjects reached the age of 25. The RAE was again significant in the prognosis. A further follow-up when the subjects are 25 years old will reveal not only what proportion of subjects are actively competing, but also if they are engaged in recreational sports, to what extent the RAE is present, and how accurately coaches can predict success.

## 1. Introduction

Lower secondary schools with a physical activity profile have become popular in Sweden, both among youngsters and local politicians. This trend has been both acclaimed and criticized [1,2,3,4,5,6,7,8,9]. Although the purpose of the school is not clearly defined, the focus seems to be on “good grades, good health and good sport results”.

Malmö Youth Sport Study (MYSS) involves two cohorts of girls and boys (*N* = 156) that attended Malmö Idrottsgrundskola (lower secondary school) between the ages of 13 and 16. After Idrottsgrundskolan, 60 percent of the pupils were admitted to a sports gymnasium (upper secondary school) between the ages of 17 and 19. The aim of this longitudinal and multidisciplinary study is to increase knowledge of how to create an activity that is inclusive and enables a commitment to sport and a lifelong interest in physical activity and sport (a health perspective) and at the same time a successful talent identification and talent development (an elite perspective). The project was formed in 2012 by researchers from the Universities of Halmstad, Lund, and Malmö. The researchers represent different disciplines and faculties, and hence different angles related to the research questions. MYSS is based on baseline data and a three-year, a six-year, and (eventually) a twelve-year follow-up. The project includes social, psychological, and physiological studies, as well as gender aspects [10,11,12,13,14,15,16,17,18,19,20,21]. In this article, talent development after secondary school (the three-year follow-up) and upper secondary school (the six-year follow-up) are analyzed, as well as an additional, *hypothetical* measure, based on information from the athletes’ trainers, predicting the category the pupils can be expected to belong to at the age of 25. The analysis is based on the variable ACHIEVE (Achieved set goals for elite effort). Additional variables related to ACHIEVE that have been analyzed are sex/gender, socio-economic position of the parents, ethnicity, sports capital, completed secondary sports school, and quartile of birth [22].

## 2. Material and Methods

### 2.1. Participants

Every year 48 boys and 30 girls are admitted to Idrottsgrundskolan. The reason for the difference in numbers is a historical one. The school started out of the Youth Academy of Malmö Football Club (MFF), so every male player from this club was, and is, attending the school. Eventually, more individuals representing different sports were admitted. In the years of the two cohorts of the Malmö Youth Sport Study, twelve sports were represented: football, swimming, diving, tennis, squash, figure skating, badminton, track and field, ice hockey, basketball, floor boll, and gymnastics. However, every year at least 20 football-playing boys representing Malmö FF are admitted. The MYSS group of two cohorts thus contains 44 football-playing boys and 18 football-playing girls (in all 62 out of 156, i.e., 40%).

To become an elite athlete, you must be talented. To develop your talent, first, it has to be identified. Over two consecutive years, 593 girls and boys applied for the sports school at the age of 13. One out of four—156—were admitted. Their marks from the sixth grade were not used in the selection process, only their achieved sports results and a forecast of future sports development. The selection was made by local or regional representatives of the different sports federations together with the school officials. The pupils were accepted on the basis of a favorable prognosis of becoming elite athletes. In the fierce competition, everyone admitted was thus forecasted to achieve results on a national and/or international level.

### 2.2. Measures

The MYSS project includes social, psychological, and physiological studies, as well as gender aspects. The sociological part of the study consists of two questionnaires, one for pupils and one for parents. Only the parent questionnaire was used in this study. It contains questions about with whom the child lives and the family form of housing; in which country the parents were born; the number of siblings; if the family had access to a car, boat (with possibilities to sleep in), summer house, or camper; the education and occupation of the parents; as well as the family’s sports experience.

### 2.3. Procedure

To measure the sports results objectively, the variable of ACHIEVE was used in the study. To “Achieve set goals for elite effort” means that you accomplish what you set out to do in terms of sports results when you started your education. The trainers and/or representatives of the district organizations were interviewed five times during 2015–2020 (in all 62 interviews). The pupils were divided on the basis of ACHIEVE, which grouped the pupils into a dropout group, a group not achieving set goals, a group performing at a district level, and one group performing at a national/international level. The individuals in the dropout group abandoned their sports efforts but they continued to attend the Malmö Idrottsgrundskola. Three out of four groups are empirically visible and normatively neutral. The second group (which did not achieve set goals) is, however, built by subjective assessments. However, this group can be objectively defined in another sense: it contains those who do not belong to one of the other three groups. For practical as well as statistical reasons, in a second step, the dropout group and the group not achieving set goals were merged into one group. The three groups are hereafter called “under district”, “district”, and “elite”. In the statistical analysis, the groups were assigned 1, 2, and 3, respectively. In the analysis, we used the ACHIEVE category obtained three years after admission to the school and six years after admission when the subjects had finished upper secondary school. In addition, we asked the trainers if they could give a hypothetical “informed guess” as to which ACHIEVE group every individual will likely belong to at the age of 25 (prognosis).

### 2.4. Data Analysis Plan

We analyzed the ACHIEVE variable in relation to the socio-economic position of the parents, ethnicity, sports capital, progression to an upper secondary sports school, sex, and the relative age effect. Information about the parents’ level of education, occupation, and country of birth was obtained from the questionnaire and, when possible, compared to statistics for the population of Malmö. We scored education from 0 to 3 (0: not completed basic education, 1: completed basic education, 2: completed upper secondary school, 3: college or university exam). The scores for the parents were added. Occupation was categorized according to the socio-economic divisions (SEI) from Statistics Sweden (Mis 1982:4) and scored from 1 to 3, with 1 being the least-qualified occupation. The SEI is based mainly on occupations and is divided into workers, civil servants, and business owners. The scores for the two parents were added. 

Three questions in the parental questionnaire covered the families’ experience with sports. First, whether the parents themselves had been actively engaged in sports, secondly, whether they had served as coaches/leaders in a sports club, and thirdly, whether any sister or brother of the pupil had been active in sports. A score was constructed in which one point was awarded for each of the items, giving a maximum of 5 points (both parents previously active and also serving as coaches/leaders and at least one active sibling). 

Information about progression to upper secondary sports school was obtained from the trainers and school officials. 

Finally, we analyzed ACHIEVE in relation to age on admission to the sports school. It is well known that recruitment by elite sports groups of youngsters is weighted to the older subjects within a given age range, a phenomenon known as the relative age effect (RAE). We have previously shown that the RAE is a determining factor for admission to Idrottsgrundskolan and that this is related to physical maturity [10]. RAE was analyzed relative to birth quarter as well as to age as a continuous variable.

Statistical analysis was mainly performed by non-parametric methods as specified in each case. 

## 3. Results

As the pupils were all accepted on the basis of a favorable prognosis of becoming elite athletes on a national and/or international level, there was only one baseline position for the pupils. After three years this was only confirmed for 28% of the pupils (Table 1); after six years for 26%. The grouping according to ACHIEVE thus included:

When groups were assigned 1–3, the mean value (±SD) for the ACHIEVE variable was 1.96 ± 0.78 after three years and 1.78 ± 0.84 after six years. The decline was statistically significant by Wilcoxon’s signed-rank test (*p* = 0.04).

Transitions between the ACHIEVE groups are shown in Figure 1. The members of the elite group who lived up to expectations on admission to the school, by and large, kept their position over time. Out of 44 pupils, after three years 31 were elite athletes, and after six years a further 10 advanced from the district group. The middle group lost 40% of its members at three years to the below-district group, whereas ten pupils advanced to the elite group. Twelve percent of the youngsters in the below-district group advanced to the middle group; none advanced to the elite group.

The informed guesses made by the coaches indicate that a substantial number of elite athletes will abandon their efforts before the age of 25. An equal number in the district group is predicted to be lost to the below-district group, whereas only three athletes are expected to advance from the district to the elite level. 

The flow between the groups is clearly seen in Figure 2, showing changes between the three-year follow-ups and the prognosis for the age of 25.

The results of the follow-ups can be summarized as follows:Two-thirds (65.3%) of all pupils belonged to the same group at the ages of 16 and 19.The elite group lost and gained roughly the same number of individuals from and to the district group.The district group lost 40% of its members to the under-district level group.There were no “late bloomers”. Among those belonging to the under-district group after three years, no one made national/international results up to the age of 19.

### 3.1. Correlation between ACHIEVE and Other Variables

According to their trainers, 24 out of the 44 pupils will remain elite after twelve years of elite effort. Given the grouping of the pupils at the three-year follow-ups and the six-year follow-ups, are there any correlations, positive or negative, between ACHIEVE and the other variables? 

### 3.2. The Socio-Economic Positions of the Parents 

We have previously shown that the socio-economic position dominating the parental group is that of the upper-middle class, characterized by parents who were largely born in Sweden, have a high educational and occupational level, and are occupationally dominated by senior officials and employers who live in well-defined families in detached or terraced houses. These characteristics strongly influenced who was admitted into the Idrottsgrundskolan [20]. The only sociological variable found to be related to ACHIEVE in this analysis was the occupation of the parents. The parents of children in the under-district group consistently had the most advanced jobs and the parents of the district group the least advanced. The differences were, however, small, the median score for occupation being 4 for the elite district as well as the below-district groups at both 3 and 6 years with interquartile ranges (IQR) of 4–5, 4–4, and 4–5, respectively, both at three and at six years (*p* = 0.02 by Kruskal–Wallis test). We found no relationship between the parents’ level of education and ACHIEVE. 

### 3.3. Ethnicity

Of the MYSS mothers, 82 percent were born in Sweden [20]. Among the others, 22 different countries were stated, albeit no country more than three times. The fathers born in Sweden also constituted 82 percent. Among the rest, 23 different countries were stated, none, however, more than three times. The proportion of children (aged 0–17) in Malmo with a foreign background in 2008 was 56% [23]. Our results thus show that neither the foreign-born mothers nor fathers represent the composition of all foreign-born Malmö female citizens. Malmö’s population represents 183 different foreign countries. In all, the foreign-born MYSS parents were not representative of the foreign-born population of Malmö—neither proportionally, nor regarding place of birth. Thus, ethnicity is one of the variables that formed the composition of the group admitted to the Idrottsgrundskolan. It did not, however, affect the grouping of ACHIEVE between baseline and the three-year or six-year follow-ups.

### 3.4. Sports Capital

The MYSS parents and siblings possess an extensive sports capital, the median score being 3 with an IQR of 3–4. Only eight pupils out of the 153 we have information on have parents that lack experience with sports. Seven out of these have one or more siblings that have been active in a sports club, and some of them still are. Thus, only one child had a family that lacked sports experience. The amount of sports capital possessed by the family was also one of the variables that formed the composition of the group admitted to the Idrottsgrundskolan. It did not, however, affect the grouping of ACHIEVE between baseline and the three-year or six-year follow-ups.

### 3.5. Completed Upper Secondary Sport School

A large proportion of the subjects in this study might be expected to apply for Idrottsgymnasiet (the upper secondary sports school) after having completed the Idrottsgrundskolan. In fact, 60% applied—31 out of 60 girls (52%) and 63 out of 96 boys (66%). 

To qualify that assumption, this should be true for those belonging to the ACHIEVE elite group and district level group. The better sports results during Idrottsgrundskolan, the more likely they applied for and were admitted to the upper secondary sports school. On the other hand, after Idrottsgrundskolan, close to four out of ten pupils had ended their elite efforts. This seems to have affected their way of choosing the next school form, as a majority from the below-district group did not attend an upper secondary sports school, whereas essentially the same proportion of subjects in the district and the elite groups did attend an upper secondary sports school (Figure 3).

### 3.6. Sex

There are differences between girls and boys regarding ACHIEVE. The transitions between the groups for boys are shown in Figure 4. Two-thirds of the elite group continue to compete at the same level after six years. The elite group loses one-third of its members to the district group and an approximately equal number are recruited from the district group. About 40% of the district group continue at the same level after six years and the same proportion drops to the below district group. The latter group appears fairly stable, retaining more than 80% of its members while gaining a substantial number from the district group.

Among the girls, the loss from the elite group from three to six years is somewhat smaller than for the boys (Figure 5) but only two girls make the transition from the district to the elite group. The elite group of girls has, in contrast to the corresponding group of boys, thus shrunk by 17%. The district group retains about half of its members, whereas an almost equal number are lost to the below-district group. The latter group retains all its original members and, therefore, grows considerably.

The changes in ACHIEVE are summarized in Table 2. Very few girls travel upward. However, a larger proportion of girls than boys are in the elite group after three years and retain that position after six years. Boys travel upward to a larger extent. An approximately equal proportion of girls and boys (25.0 and 24.0%, respectively), made a downward journey. More girls traveled downward and more boys traveled upward. 

The sex differences can be summarized as follows:After three years, the proportion of girls in the elite group is larger than for the boys (38 vs. 22%).After six years, the number of girls in the elite group has declined, but proportionally the elite group of girls is still larger than the boys (32% vs. 23%).All girls in the under-district group after three years are still in the under-district group after six years (6 out of 34 boys moved into the district group).Almost half of the pupils belong to the under-district group after six years. This is the case both for girls and boys.There were no “late bloomers”, neither among the boys nor the girls. All girls in the under-district group after three years were still in the under-district group after six years. Six boys were upgraded from the under-district level to the district level.Thus, attending the sports gymnasium (upper secondary school) does not improve the development of the girls’ sports results.

### 3.7. Relative Age Effect

The distribution of the birth quarter of the pupils admitted to Idrottsgrundskolan is shown in Figure 6. The distribution is skewed (*p* < 0.05 by *Χ*^2^ test) with very few subjects born in the fourth quarter. 

The relative age effect was significantly influencing not only the admission to the Idrottsgrundskolan but also the forming of the ACHIEVE groups as well as the reshaping of them during the three years spent in the school (Table 3). The same was not the case during the period at the upper secondary sports school, but in the prognosis for their next six years a significant pattern once again appears. Even if the difference is small, the effect is considered moderate as assessed by Hedge’s g (0.60 at 3 years and 0.73 for the prognosis).

## 4. Discussion

In this study, we analyze sports development in a lower secondary sports school. The relationship between school and sport is seldom problematized. Many claims have been made about the mutual benefits of school results and sports results [24]. Most people think of it in a win-win manner: school is good for sport and sport is good for the school. In society as well as in research materials, there is a general understanding that PA is good [25,26,27]. Research conclusions have been drawn that PA, beyond the physiological effects, a supportive environment and context can lead to persistence, discipline, commitment, wellbeing, self-esteem, anti-depression, stress reduction, the feeling of belonging, networks, status, and an increased ability to concentrate [25]. Increasing weekly PA over nine years has been associated with improved academic achievement in boys [28,29]. All these effects are in the favor of good school results. At the same time, sports schools are believed to help the pupil´s sports careers by strengthening their international competition capacity, facilitating dual careers, strengthening the professionalization of sports, and supporting public health [2,4,30].

A large amount of knowledge combined with good subject-specific knowledge, as well as characteristics such as disciplined handling of studying, concentration capacity, and a talent for long-term thinking, are all assets that are regarded as being in favor of sports development. At the same time, scientific results claim that intensive physical activity works in the favor of good school results.

Based on a statistical analysis, we have discussed the variables of ACHIEVE to understand the differences in talent development for two cohorts of athletes attending Idrottsgrundskolan in Malmö. Three sets of conclusions are drawn. The first one relates to changes in the groupings between baseline, at three years, at six years, and at hypothetically twelve years of development. The second is related to the effects of societal variables on the outcomes. The third one is related to the biological variables affecting the outcomes. 

The overall changes in the ACHIEVE groups could be described as a gradual polarization: a group where every pupil was defined as elite at baseline, became a large middle group after three years, and then a large under-district group after six years, at the same time as the elite group became notably stable but somewhat smaller. This process continues—hypothetically—for up to twelve years. Either there was an incorrect prognosis at baseline in seven out of ten cases, or things happened in seven out of ten cases related to, for instance, injuries, effects on the sports development on behalf of the athlete or his/her trainer, or social factors. 

The polarization can be understood in relation to a number of factors that influence talent development during lower and upper secondary school, including school-related factors, sports-related factors, and social factors. Some factors are logically linked to the talent-development process, for example, physical and psychological factors [11,15,17]. Another such factor, detectable through the relative age effect is physical maturity [10]. As very few individuals became internationally successful, one factor influencing the outcome is an unrealistic optimism in the younger age groups, which is often eventually replaced by a more sober judgement about the meaning as well as the effects, of the commitment. The competition logic pushes the athletes in different directions; both wins and losses have to be handled. In addition, the impact of injuries is obvious, as is the influence—good or bad—of trainers and coaches. The role of coaches has been demonstrated in a study by Witkowski et al. [31].

If the outcome in terms of ACHIEVE, from the point of view of a lower secondary sports school, is to be regarded as either a successful or unsuccessful development, it can be discussed in the form of “whether the glass is half full or half empty”. On the one hand, it is difficult to claim that a special sports school form is needed to produce elite athletes on a national and/or international level if only 28 percent had succeeded when they entered the Idrottsgrundskolan at the age of 16. On the other hand, 45 pupils from one lower secondary school in the third-largest city in a small country, having achieved sports results at a national and/or international level at the age of 16, can be argued as being a very successful development. Out of these—hypothetically—24 could remain elite athletes up to senior age. However, the counterargument needs to be problematized by the fact that half of these 24 elite athletes were admitted to Idrottsgrundskolan because they belonged to the most advanced and selective Football Youth Academy in Sweden, Malmö FF. They did not need Idrottsgrundskolan to acquire the amount of support—quantitatively or qualitatively—needed to become successful football players. Similarly for a number of the other half of the elite athletes. All in all, the results do not present a strong case for the conclusion that this lower secondary sports school is needed to identify and develop elite athletes on a national and/or international level. However, to draw that conclusion we still have to wait for the official twelve-year follow-ups. 

After three years of talent development and then three more years of upper secondary school, we cannot rule out that the pupils could still become elite athletes as adults. Only the dropouts seem like “lost cases”. On the other hand, behind the label “dropout” we have found that many of the boys and girls who have stopped exercising for elite sports themselves are coaching and training others, or they dropped out because of sports injuries. Nevertheless, after three years of development, one out of three has stopped doing sports or has not achieved set goals for elite efforts. After six years, half of the pupils belonged to this group.

The background factors regarded within social sciences as being of profound importance when dividing citizens into different groups, which are later affected in different ways and with different results, are age, sex/gender, social class/group (educational level and occupation), ethnicity, and geographical location/population density [32,33]. The MYSS parent questionnaire includes questions about most of these factors as well as questions about sports habits [34,35]. Sport and physical exercise habits are clearly marked socially, socioeconomically, socio-geographically, and culturally [20,34,35,36,37]. The socio-economic position dominating the parental group in this study is that of the upper-middle class, characterized by parents who were largely born in Sweden, have a high educational and occupational level, and are occupationally dominated by senior officials and employers who live in well-defined families in detached or terraced houses. Obviously, these characteristics strongly influenced who was admitted into the Idrottsgrundskolan [20]. The sports capital for the children at Idrottsgrundskolan was very large, the IQR being 3–4. This means that for 50 % of the group, both parents had a sports background (active and/or leader). However, only the occupation showed significance in relation to the dividing of the pupils into the ACHIEVE groups. One reason for not finding more significant associations is that the group was very homogeneous. It seems like the effects of socio-economic variables diminished once the school group was formed. After that, other variables must have decided who changed ACHIEVE groups. The main variables found in this study were both biologically related—sex and differences in physical maturity, known as the relative age effect (RAE). 

Differences in sports achievements between boys and girls can be related both to biological factors (sex) and to socially constructed factors (gender) [15,38]. When it comes to gender, this is discussed in length by Larneby [15]. Here we have focused on questions such as: What are the group compositions for the girls and the boys? How do they change between three and six years? How many girls and boys keep their original groups? How many girls and boys make a journey upward? How many girls and boys make a journey downward? There are differences between girls and boys regarding ACHIEVE. All girls in the under-district group after three years were still in the under-district group after six years, whereas 6 out of 34 boys moved into the district group. Thus, attending upper secondary sports school did not improve the girls’ sports results development. Very few girls traveled upward. However, a larger proportion of girls than boys were in the elite group after three years and retain that position after six years. More girls traveled downward, and more boys traveled upward, particularly if they (the boys) were born in quartiles 3 and 4. The effects of RAE are larger for the boys between three and six years. 

The relative age effect is a widely recognized effect of selection systems within competitive children and youth sports across a variety of sports and countries all over the world. RAE significantly influenced not only admission to the Idrottsgrundskolan but also the forming of the ACHIEVE groups as well as the reshaping of them during the three years spent at the school. The same was not the case during the upper secondary sports school, but in accordance with the prognosis for their next six years a significant pattern will once again appear.

The major strengths of this study include its longitudinal design and that the information has been obtained for all subjects at all time points. The limitations include the moderate number of subjects studied and the lack of diversity of the sports represented. Individual as well as team sports are included, which may entail differences. 

## 5. Conclusions

The children attending the Idrottsgrundskolan are characterized as having an upper-middle-class background and extensive sports capital. After completion of the lower secondary sports school, three groups of youngsters can be identified: those competing at a national/international level, those competing at a district level, and those having more or less abandoned their elite efforts. The elite group, thereafter, remains fairly stable, whereas the district group is progressively reduced. Few children make an upward journey, i.e., there are no late bloomers. Few social variables affect sports achievements probably owing to the selection mechanisms for admission to the school. Sex affected sports development—a larger proportion of girls than boys reached the elite group after lower and upper secondary school, but fewer girls than boys made an upward journey during the period. The relative age effect profoundly affected admission to the school as well as achievements after lower secondary school but less so after upper secondary school.

## Figures and Tables

**Figure 1 sports-10-00071-f001:**
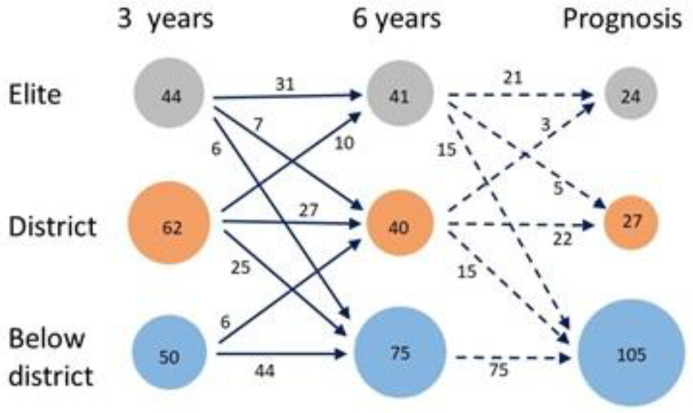
Changes from the three-year follow-ups, the six-year follow-ups, and from the last follow-up to adulthood, as predicted by the coaches (prognosis), based on the ACHIEVE variable (*N* = 156).

**Figure 2 sports-10-00071-f002:**
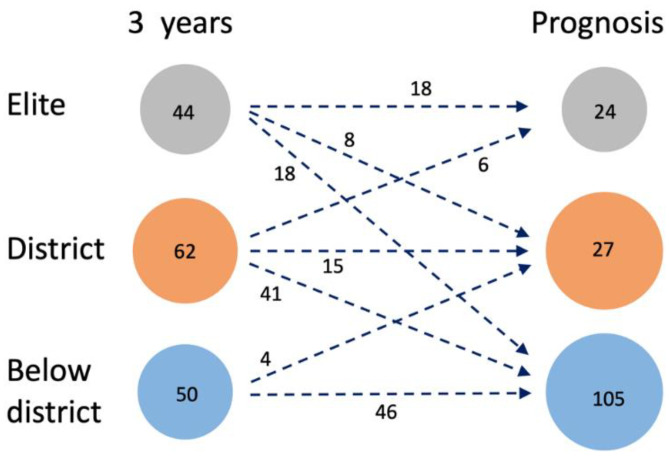
Changes from the three-year follow-ups to the prognosis for the twelve-year follow-ups, based on the ACHIEVE variable (*N* = 156).

**Figure 3 sports-10-00071-f003:**
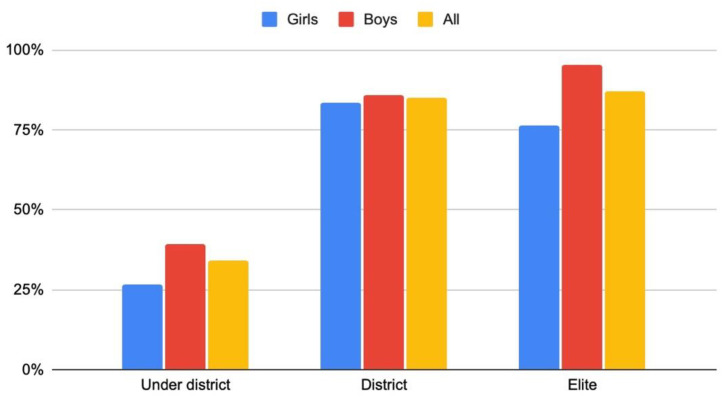
The MYSS girls and boys attending Sport Upper Secondary School (idrottsgymnasium), based on three-year follow-ups of ACHIEVE (%).

**Figure 4 sports-10-00071-f004:**
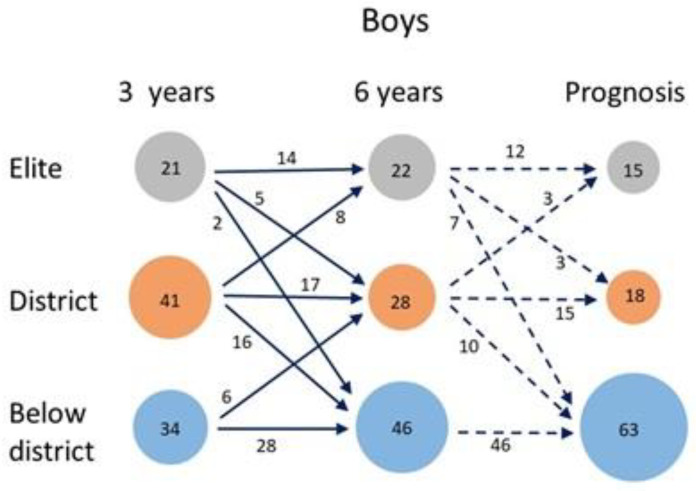
Changes from the three-year to the six-year follow-ups, based on the ACHIEVE variable for the boys (*n* = 96).

**Figure 5 sports-10-00071-f005:**
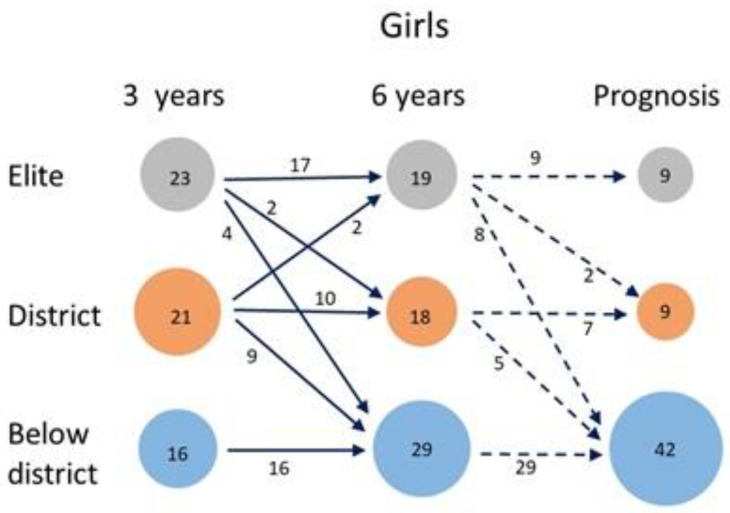
Changes from the three-year to the six-year follow-ups, based on the ACHIEVE variable for the GIRLS (*n* = 60).

**Figure 6 sports-10-00071-f006:**
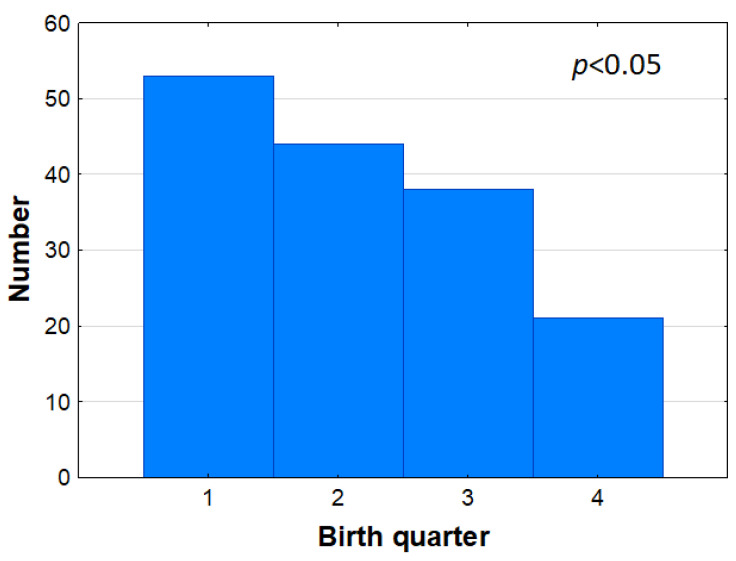
Frequency distribution of birth quarter at admission.

**Table 1 sports-10-00071-t001:** ACHIEVE divided into elite group, district group, and under district group (*N* = 156).

Group	3 Years	6 Years
Results at a national and/or International level (3)	44	41
Results at a district level (2)	62	40
Not achieved set goals (below district) (1)	50	75

**Table 2 sports-10-00071-t002:** Changes in ACHIEVE during follow-up.

**3 → 6 Years**	**Increasing**	**No Change**	**Decreasing**
All	16 (10%)	102 (65%)	38 (24%)
Boys	14 (15%)	59 (61%)	23 (24%)
Girls	2 (3%)	43 (72%)	15 (25%)
**6 Years** **→ Prog** **nosis**			
All	3 (2%)	118 (76%)	35 (22%)
Boys	3 (3%)	73 (76%)	20 (21%)
Girls	0 (0%)	45 (75%)	15 (25%)

**Table 3 sports-10-00071-t003:** ACHIEVE—RAE.

**Follow-Up**	**Mean Age at Admission (years)**	**SD**	***p* (ANOVA)**
**3 years**			
Elite	13.6 *	0.36	0.008
District	13.5	0.26	
Below district	13.4 *	0.31	
**6 years**			
Elite	13.6	0.32	0.096
District	13.4	0.33	
Below district	13.5	0.29	
**Prognosis**			
Elite	13.6 *	0.36	0.010
District	13.4 *	0.36	
Below district	13.5	0.28	

* Significant differences between groups by Tukey’s test.

## Data Availability

The data presented in this study are available on request from the corresponding author.

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
