# Peer review of "Changes in Athletic Performance in Children Attending a Secondary School with a Physical Activity Profile"

_sports, 2022, doi:10.3390/sports10050071_

Round 1
Reviewer 1 Report
Overall thought – very interesting, appropriate to journal audience, needs editing and some more information. I appreciate being able to review the manuscript.
Abstract, provide N of participants, etc., provide more result details, conclusion, and perhaps a future direction.
Keywords, perhaps talent identification, elite sport, would be better than a few listed or at least add them.
Introduction, informative to the reader.
Line 30, merge this paragraph up to line 37 or 38.
I believe breaking up 2. Material and methods into 2.1. Participants, 2.2. Measures, 2.3. Procedure, 2.4. Data analysis plan will add future readers to best follow your methods.
Line 138, 40 % to 40%.
Figures, very informative.
You go back and forth with = having spaces (n = 156) or not (p=0.04)
Should your n be N as it is the total group size?
Table 2, % all have a space
Table 3, the Tukey test results seem deceiving. What is a .1 difference? What is the variable as the title is Relative Age Effect, I assume. So the variable is age?
The calculation is a medium Cohen’s d for both differences. I understand that via using a calculator. It seems you should include a Cohen’s d or Hedges’ g (given different sample sizes). Then tell the ready what the difference means.
Line 387-388, probably do not need to relist p values.
How about a limitations sections?
After acknowledgements, the other MDPI information is required.
Supplementary Materials: The following are available online at www.mdpi.com/xxx/s1, Figure S1: title, Table S1: title, Video S1: title.
Author Contributions: For research articles with several authors, a short paragraph specifying their individual contributions must be provided. The following statements should be used “Conceptualization, X.X. and Y.Y.; methodology, X.X.; software, X.X.; validation, X.X., Y.Y. and Z.Z.; formal analysis, X.X.; investigation, X.X.; resources, X.X.; data curation, X.X.; writing—original draft preparation, X.X.; writing—review and editing, X.X.; visualization, X.X.; supervision, X.X.; project administration, X.X.; funding acquisition, Y.Y. All authors have read and agreed to the published version of the manuscript.”, please turn to the CRediT taxonomy for the term explanation. Authorship must be limited to those who have contributed substantially to the work reported.
Funding: Please add: “This research received no external funding” or “This research was funded by NAME OF FUNDER, grant number XXX” and “The APC was funded by XXX”. Check carefully that the details given are accurate and use the standard spelling of funding agency names at https://search.crossref.org/funding, any errors may affect your future funding.
Acknowledgments: In this section you can acknowledge any support given which is not covered by the author contribution or funding sections. This may include administrative and technical support, or donations in kind (e.g., materials used for experiments).
Conflicts of Interest: Declare conflicts of interest or state “The authors declare no conflict of interest.” Authors must identify and declare any personal circumstances or interest that may be perceived as inappropriately influencing the representation or interpretation of reported research results. Any role of the funders in the design of the study; in the collection, analyses or interpretation of data; in the writing of the manuscript, or in the decision to publish the results must be declared in this section. If there is no role, please state “The funders had no role in the design of the study; in the collection, analyses, or interpretation of data; in the writing of the manuscript, or in the decision to publish the results”.
Author Response
Response to the referees’ comments
Comments and Suggestions for Authors
Overall thought – very interesting, appropriate to journal audience, needs editing and some more information. I appreciate being able to review the manuscript.
Thank you!
Abstract, provide N of participants, etc., provide more result details, conclusion, and perhaps a future direction.
The abstract has been revised according to the suggestions.
Keywords, perhaps talent identification, elite sport, would be better than a few listed or at least add them.
The suggested key words have been added.
Introduction, informative to the reader.
Line 30, merge this paragraph up to line 37 or 38.
Done.
I believe breaking up 2. Material and methods into 2.1. Participants, 2.2. Measures, 2.3. Procedure, 2.4. Data analysis plan will add future readers to best follow your methods.
Done.
Line 138, 40 % to 40%.
Done.
Figures, very informative.
You go back and forth with = having spaces (n = 156) or not (p=0.04)
This should now be consistent.
Should your n be N as it is the total group size?
Correct and changed.
Table 2, % all have a space
Corrected.
Table 3, the Tukey test results seem deceiving. What is a .1 difference? What is the variable as the title is Relative Age Effect, I assume. So the variable is age?
The calculation is a medium Cohen’s d for both differences. I understand that via using a calculator. It seems you should include a Cohen’s d or Hedges’ g (given different sample sizes). Then tell the ready what the difference means.
The variable is age, used as a continuous variable, which is now specified in the table. While the differences are certainly small in absolute terms, they are nonetheless statistically significant. We have calculated the Hedge’s g and included this information in the text.
Line 387-388, probably do not need to relist p values.
Removed.
How about a limitations sections?
This has been included.
After acknowledgements, the other MDPI information is required.
Supplementary Materials: The following are available online at www.mdpi.com/xxx/s1, Figure S1: title, Table S1: title, Video S1: title.
Author Contributions: For research articles with several authors, a short paragraph specifying their individual contributions must be provided. The following statements should be used “Conceptualization, X.X. and Y.Y.; methodology, X.X.; software, X.X.; validation, X.X., Y.Y. and Z.Z.; formal analysis, X.X.; investigation, X.X.; resources, X.X.; data curation, X.X.; writing—original draft preparation, X.X.; writing—review and editing, X.X.; visualization, X.X.; supervision, X.X.; project administration, X.X.; funding acquisition, Y.Y. All authors have read and agreed to the published version of the manuscript.”, please turn to the CRediT taxonomy for the term explanation. Authorship must be limited to those who have contributed substantially to the work reported.
Funding: Please add: “This research received no external funding” or “This research was funded by NAME OF FUNDER, grant number XXX” and “The APC was funded by XXX”. Check carefully that the details given are accurate and use the standard spelling of funding agency names at https://search.crossref.org/funding, any errors may affect your future funding.
Acknowledgments: In this section you can acknowledge any support given which is not covered by the author contribution or funding sections. This may include administrative and technical support, or donations in kind (e.g., materials used for experiments).
Conflicts of Interest: Declare conflicts of interest or state “The authors declare no conflict of interest.” Authors must identify and declare any personal circumstances or interest that may be perceived as inappropriately influencing the representation or interpretation of reported research results. Any role of the funders in the design of the study; in the collection, analyses or interpretation of data; in the writing of the manuscript, or in the decision to publish the results must be declared in this section. If there is no role, please state “The funders had no role in the design of the study; in the collection, analyses, or interpretation of data; in the writing of the manuscript, or in the decision to publish the results”.
This information has been included.
Comments and Suggestions for Authors
At the outset, I would like to thank you for the opportunity to review your work.
Thank you for your effort!
I evaluate the article positively, but in some places corrections are necessary. 1. The abstract is too general and does not fulfill its role.
Please re-edit the abstract taking into account the background and purpose of the study, the methods section, results and conclusions in a form that the reader can understand at first reading.
The abstract has been revised according to the suggestions.
- I did not find in the text the approval number of the test results for publication by the Local Ethics Committee.
This should be completed in the method section.
Information about ethics approval has been included as suggested.
In the discussion, please refer to the important role of the trainer in educating young people.
Please refer to the article: Witkowski K, Proskura P, Piepiora P. The role of a combat sport coach in the education of youth - a reference to the traditional standards and perception of understanding the role of sport in life of an individual and society. Arch Budo Sci Martial Art Extreme Sport 2016; 12: 123-130.
Thank you for pointing out this interesting article. It is now referenced in the text.
Reviewer 2 Report
At the outset, I would like to thank you for the opportunity to review your work.I evaluate the article positively, but in some places corrections are necessary. 1. The abstract is too general and does not fulfill its role.
Please re-edit the abstract taking into account the background and purpose of the study, the methods section, results and conclusions in a form that the reader can understand at first reading. 2. I did not find in the text the approval number of the test results for publication by the Local Ethics Committee.
This should be completed in the method section.
3. In the discussion, please refer to the important role of the trainer in educating young people.
Please refer to the article: Witkowski K, Proskura P, Piepiora P. The role of a combat sport coach in the education of youth - a reference to the traditional standards and perception of understanding the role of sport in life of an individual and society. Arch Budo Sci Martial Art Extreme Sport 2016; 12: 123-130.
Author Response

(The authors gave the same response as above.)

Round 2
Reviewer 1 Report
I believe the authors made all corrections and addressed all suggestions. Thank you.
I clicked accept after minor revisions as the authors need to number in their manuscript the references. I am sure everyone understands this need. That is a good idea until the almost end.